# Morphology-Controlled Synthesis of ZnO Nanostructures for Caffeine Degradation and *Escherichia coli* Inactivation in Water

Shaila Thakur [1], Sudarsan Neogi [1] and Ajay K. Ray [2,*]

[1]  Department of Chemical Engineering, Indian Institute of Technology, Kharagpur 721302, India; shaila.thakur@hotmail.com (S.T.); sneogi@che.iitkgp.ernet.in (S.N.)
[2]  Department of Chemical and Biochemical Engineering, Western University, London, ON N6A 3K7, Canada
*   Correspondence: aray@eng.uwo.ca

**Abstract:** Photocatalytic and antibacterial activity of nanoparticles are strongly governed by their morphology. By varying the type of solvent used, one can obtain different shapes of ZnO nanoparticles and tune the amount of reactive oxygen species (ROS) and metal ion ($Zn^{2+}$) generation, which in turn dictates their activity. ZnO nanostructures were fabricated via facile wet chemical method by varying the type of solvents. Solar light assisted photocatalytic degradation of caffeine and antibacterial activity against *E. coli* were examined in presence ZnO nanostructures. In addition to an elaborate nanoparticle characterization, adsorption and kinetic experiments were performed to determine the ability of nanostructures to degrade caffeine. Zone of inhibition, time kill assay and electron microscopy imaging were carried out to assess the antibacterial activity. Experimental findings indicate that ZnO nanospheres generated maximum ROS and $Zn^{2+}$ ions followed by ZnO nanopetals and ZnO nanorods. As a result, ZnO nanospheres exhibited highest degradation of caffeine as well as killing of *E. coli*. While ROS is mainly responsible for the photocatalytic activity of nanostructures, their antibacterial activity is mostly due to the combination of ROS, metal ion, physical attrition and cell internalization.

**Keywords:** zinc oxide; nanosphere; nanorod; nanopetal; photocatalytic; antibacterial; caffeine; reactive oxygen species (ROS); degradation; pathogen

## 1. Introduction

Caffeine is one of the most abundant xenobiotics that causes water pollution due to its high daily consumption across the globe [1]. Apart from food industry, caffeine is also extensively used in the pharmaceutical industry [2,3]. Another pollutant found in wastewater effluents that causes serious health concerns are pathogenic microorganisms. *Escherichia coli* (*E. coli*) is a common pathogen found in aquatic environment and is known to cause several enteric diseases even at low concentrations [4]. Hence, it is imperative to treat these detrimental effluents before they can be discharged to water bodies. The conventional techniques employed by researchers for wastewater treatment are easy to handle and in most cases reusable [5]. However, most of the processes such as activated sludge-based technique, ion-exchange and coagulation have low treatment efficiencies and they are not cost effective in the long run. Even the advanced techniques such as membrane filtration faces a serious drawback of fouling that increases energy consumption and lowers the separation efficiency. To overcome these obstacles, nanotechnology offers a versatile and promising solution for the degradation of organic matter and elimination of microbes from wastewater in a cost-effective way.

A variety of nanomaterials have been investigated as photocatalysts for treating emerging contaminants from wastewater [6,7]. ZnO was selected for this study because of its low cost, simple synthesis technique and it can safely be used as antibacterial agent in food products [8–11]. It has been reported that ZnO nanoparticles can degrade a variety of contaminants and are effective against a wide range of bacteria but ZnO morphology

largely governs its photocatalytic and antibacterial properties [12–16]. A comparison of how different shape of ZnO nanoparticles affect its photocatalytic and antibacterial activity (against *E. coli*) is shown in Table 1. Despite the fact that morphology governs the photocatalytic activity of ZnO, there has been limited research to study the shape dependent photocatalytic and antibacterial activity of nanomaterials. A fundamental understanding of difference in activities arising from different nanoparticle morphologies and their mechanism of action is still unknown. Moreover, most of the previous studies have utilized UV irradiated nanoparticles for the photocatalytic degradation of caffeine [17–19].

**Table 1.** Comparison of photocatalytic and antibacterial activity of different shapes of ZnO nanoparticles.

| | Particle Size (nm) | Model Pollutant | Photocatalytic Activity (% Degradation) | Maximum %Reduction in *E. coli* Growth | Ref |
|---|---|---|---|---|---|
| ZnO sphere | 9.6–25.5 | Methylene blue (MB) | 82.1% at 180 min UV exposure | 69.2% at 100 μg/mL | [20] |
| | 133.7–260.2 | MB | 18% and 29% after 1 h UV exposure (1 g/L) | 30 and 35% after 1 h exposure (1 g/L) | [21] |
| | 53.99 | MB | 95.45% after 180 min | Not reported | [22] |
| | 4.35 | 4-nitrophenol | 78% in 100 min | 85% in 5 h at 100 μg/mL | [13] |
| | 65.00 | Acid Orange 74 | 80% after 80 min | 99.93% at 20 min at 20 ppm NP | [23] |
| ZnO petals | 214.38 × 178.22 | MB | 96.52 after 180 min | Not reported | [22] |
| | 45.0 | Acid Orange 74 | 90% after 80 min | 99.97% after 20 min at 20 ppm NP | [23] |
| | (1.41–1.8) × (0.33–0.4) | methylene blue and Congo red | 81% for CR and 67% for MB after 80 min | 90% for 150 mg/mL after 6 h | [24] |
| ZnO rod | 155.0 | MB | 87.12 after 180 min | Not reported | [22] |
| | 20.0 | Orange II | 100% after 150 minsolar irradiation with 1 mg/mL | 100% in >3 h (1 mg/mL) | [25] |
| | 76.0 | Acid Orange 74 | 70% after 80 min | 99.8% after 20 min t 20 ppm NP | [23] |

This study aims at determining the photocatalytic dependence of different nanostructures using solar light which is relatively inexpensive compared to the UV light. There are studies that individually report different nanostructures for their photocatalytic and antibacterial activity, but to the best of our knowledge, this is the first report to study the comparative solvent induced morphology-dependent photocatalytic and antibacterial mechanism of action of ZnO nanostructures. The goals of this study are: (i) To investigate the adsorption of caffeine on different morphology of nanoparticles (in dark) (ii) To evaluate the effect of initial caffeine concentration, amount of nanostructures and solar light intensity on the rate of caffeine degradation and estimate their kinetic rate constants, and (iii) study the antibacterial activity of ZnO nanostructures on *E. coli*.

## 2. Results and Discussion

### 2.1. Nanoparticle Characterization

2.1.1. Electron Microscopy

From the SEM images (Figure 1a–c), ZnO appears as well-defined nanospheres, nanorods and nanopetals under the influence of different solvents. PEG400 gives rise to spherical morphology; for water, we obtain petals attached to each other and for toluene, we obtain nanorod like structure. The morphological parameters are presented in Table 2. From the TEM images (Figure 1d–f), it can be seen that nanospherical ZnO are the smallest of all structures with an average diameter of 10.18 nm. Nanorods exhibited a mean width of 157 nm and a length of 1.43 μm while nanopetals have an average thickness of 31.85 nm. The EDX analysis of nanoparticles is shown in the Supplementary Information (Table S1). It can be seen that the elemental composition does not change significantly with the change in ZnO morphology.

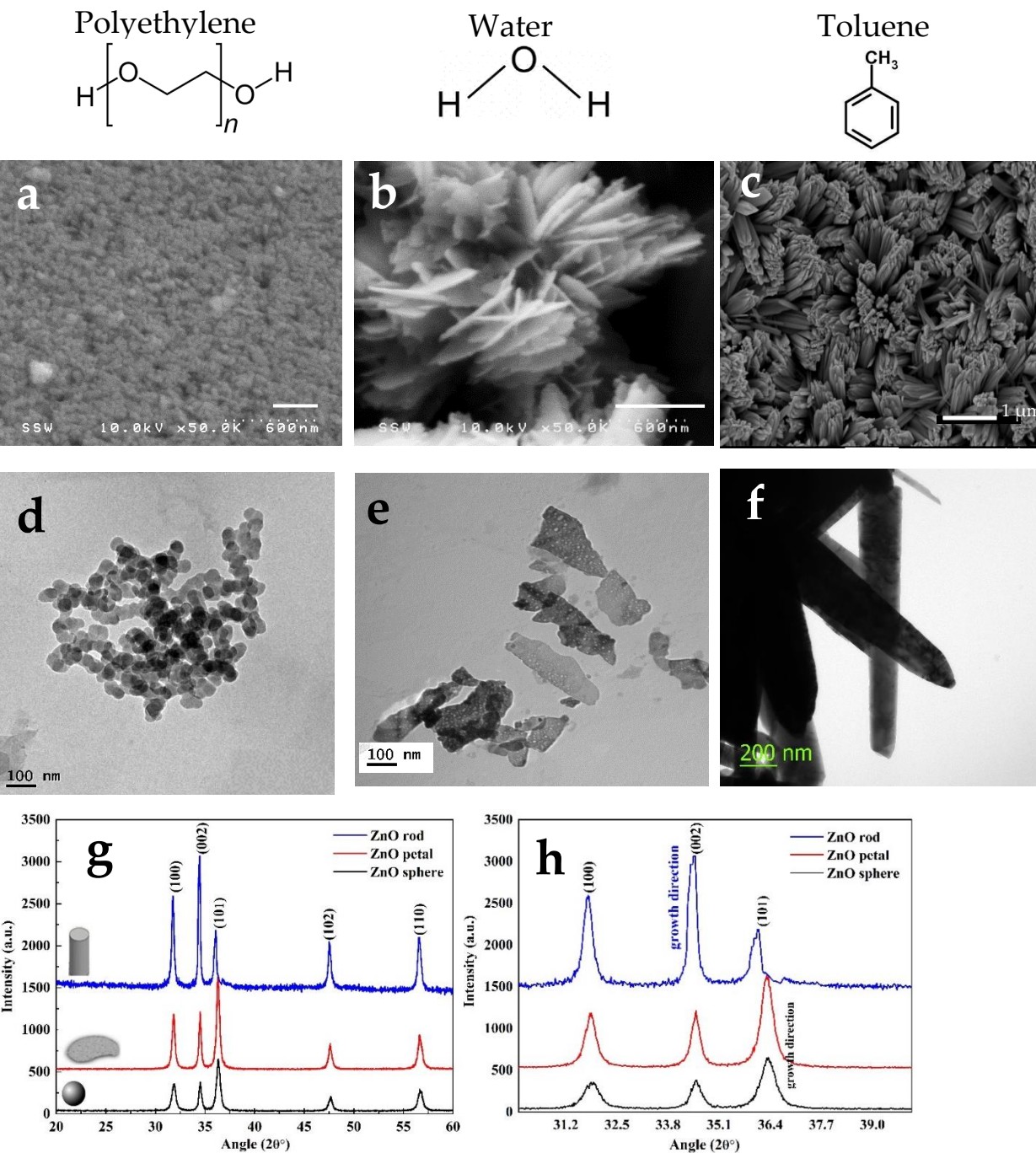

**Figure 1.** SEM (**a–c**) and TEM (**d–f**) images of ZnO nanomaterials synthesized in different solvents (**a,d**) PEG400 (**b,e**) water (**c,f**) toluene. (**g**) XRD spectra of different ZnO shapes (**h**) XRD spectra in the 2θ range 30–40° showing the crystal growth direction.

**Table 2.** Characterization of ZnO nanostructures.

| Sample | Average Particle Diameter (nm) (TEM) | BET Specific Surface Area (m²/g) | Total Pore Volume (cc/g) | Average Pore Diameter (nm) | Point of Zero Charge |
|---|---|---|---|---|---|
| ZnO sphere | 10.18 | 92.22 | 0.15 | 6.64 | 4.90 |
| ZnO petal | 31.85 (petal thickness) | 12.02 | 0.03 | 10.70 | 6.00 |
| ZnO rod | 157.00 (diameter) | 6.60 | 0.017 | 10.26 | 6.80 |

### 2.1.2. BET

The BET parameters for different morphologies of ZnO nanoparticles is given in Table 2. According to these results, the synthesized nanostructures have a mesoporous nature [26]. ZnO sphere had the largest specific surface area, highest pore volume and lowest average pore diameter. ZnO petals and rods has different specific surface areas and total pore volume but the average pore diameters were similar. These parameters are used to determine the effectiveness of nanoparticles.

### 2.1.3. X-ray Diffraction (XRD)

Figure 1g,h shows the XRD spectra of ZnO nanoparticles prepared in different solvents. The peaks at 31.85°, 34.52°, 36.23°, 47.50°, 56.50°, 62.9°, 66.29°, 67.92° and 68.82° indicate wurtzite hexagonal ZnO structure (JCPDS 89-1397). The preferred growth direction for ZnO were (101) for spheres and petals and (002) for ZnO rods. It can be seen that the (101) peak for ZnO nanorods is shifted towards lower diffraction angle. This may be attributed to the change in growth direction from (101) to (002) thereby reducing the strain along (101) direction.

### 2.1.4. Zeta Potential and $Zn^{2+}$ Generation

The point of zero charge is indicative of the net charge of particles in the solution. A plot of zeta potential for different morphologies of ZnO is presented in Figure 2a. It was observed that the point of zero charge ($pH_{ZC}$) varied for different nanostructures. The $pH_{ZC}$ for ZnO spheres was ~4.9 (4.86) while it shifted to 6.0 for petals and 6.8 for rods. The lower $pH_{ZC}$ for spheres indicate better stability of spheres compared to other nanostructures at neutral pH. The lower $pH_{ZC}$ of ZnO spheres is due to the presence of PEG molecules on ZnO nanoparticles. It has been shown in literature that the presence of carbonates shifts the peak in the direction of lower pH [27]. Different $pH_{ZC}$ may be a result of differences in surface energy arising from the shape controlling process. Different surface energy of nanoparticle shapes leads to dissimilar adsorption of protons and hydroxyl ions on nanoparticles [28].

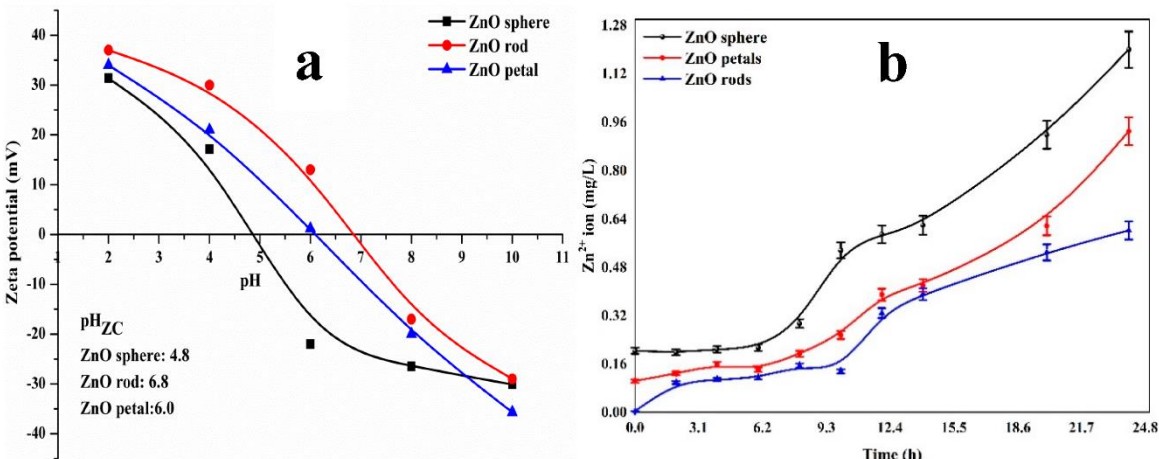

**Figure 2.** *Cont.*



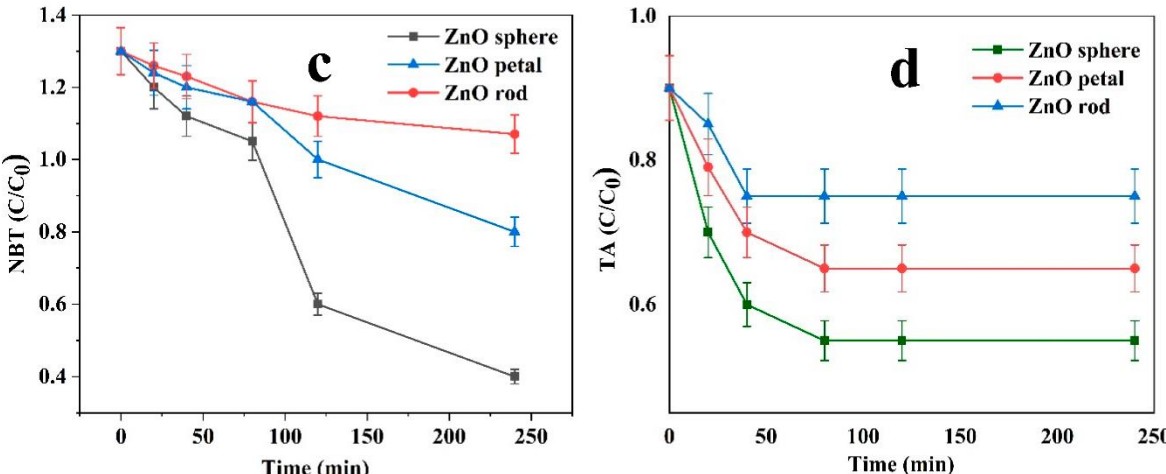

**Figure 2.** (**a**) Zeta potential and (**b**) $Zn^{2+}$ release curves of ZnO nanostructures synthesized using different solvents. Degradation kinetics of (**c**) NBT and (**d**) TA in presence of ZnO nanostructures for the quantification of superoxide ($\bullet O_2^-$) and hydroxyl radicals ($\bullet OH$), respectively.

The $Zn^{2+}$ release profile is presented in Figure 2b. The ion release rates followed an order ZnO spheres > ZnO petals > ZnO rods.

### 2.1.5. Generation of Hydroxyl (OH$\bullet$) and Superoxide Radicals ($\bullet O_2^-$)

The formation of superoxide radicals by ZnO nanostructures was analysed by nitroblue tetrazolium (NBT) assay and the plots are shown in Figure 2c. The reduction in absorbance of NBT with time was recorded and correlated with the concentration using standard calibaration curve. The extent of degradation can be directly correlated to the extent of ROS generatio which follows a simlar trend like metal ion release: ZnO spheres > ZnO petals> ZnO rods.

Terephthalic acid degradation is a commonly used technique for the detection of hydroxyl radicals [29,30]. Terephthalic acid does not show any fluorescence peak but the product of terephthalic acid and hydroxyl radicals (2-hydroxyterephthalic acid) shows a characteristic fluorescent peak at 425 nm. The degradation rate of terephthalic acid by hydroxyl radicals is plotted in Figure 2d. It can be noted that the rate of generation of OH$\bullet$ is maximum in the first 40 min after which it slows down and stops. The fluorescence spectra of terephthalic acid in absence and presence of nanoparticles after 40 min is presented in the Supplementary Information (Figure S1). The intensity of fluorescent peak is indicative of the amount of OH$\bullet$ generated by the nanostructures and thus it is evident that ZnO sphere produces maximum number of OH$\bullet$ followed by ZnO petals and ZnO rods. This is similar to the trend showed by $\bullet O_2^-$.

### 2.2. Mechanism of Morphology Change with the Choice of Solvent

The main factors that lead to change in nanoparticle morphology are: seeding and growth direction, hydrolysis and solvent templating [31–33]. Spherical nanoparticles were formed in case of PEG because glycol chains surround $Zn^{2+}$ ions and inhibit particle agglomeration. Moreover, PEG consists of two polar hydroxyl groups that bind to (002) plane (*c* direction) and promote growth along the (101) plane thus leading to spherical morphology [31]. On the contrary when water is used as a solvent, the rate of hydrolysis increases and the amount of ($[Zn(OH)_4]^{2-}$) seeds increases. To reduce the overall surface energy of the system, petal like morphology is formed that attach to each other in a "template-like" fashion so that the system moves towards a more energetically favored low energy state [31]. Rod like ZnO nanostructures are formed when the crystals selectively grow along normal of the (002) planes. For non-polar aromatic molecules such as toluene, solvent attachment to the polar (002) plane is ineffective which directs the formation of ZnO along the perpendicular (002) plane giving rise to nanorods [31].

*2.3. Caffeine Degradation*

2.3.1. Adsorption Study

A calibration curve for caffeine has been shown in the Supplementary Information (Figure S2). Figure 3 shows that the equilibrium adsorption capacity ($q_e$) increased with the increase in initial concentration of caffeine. At high caffeine concentrations, the adsorption sites get saturated and $q_e$ either remains constant or decreases due to desorption. The increase in adsorption with the increase in caffeine concentration can be attributed to the high concertation gradient which promoted the mass transfer of aqueous caffeine towards solid ZnO nanostructures. We also studied the time dependence of adsorption process and found that the adsorption of caffeine on ZnO nanostructures was a two-stage process and it reached equilibrium in almost 45 min (unaffected by the concentrations of ZnO and caffeine and type of nanoparticles). The first 20 min witnessed a rapid adsorption followed by a slower phase of 25 min of gradual uptake. A large number of unoccupied adsorption sites on ZnO nanostructures promoted an initial rapid adsorption and the rate slowed down when most of these sites were occupied.

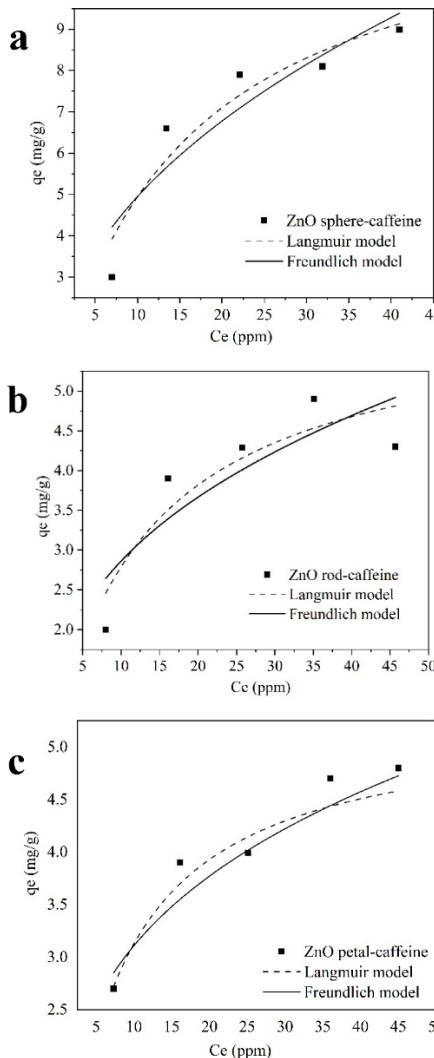

**Figure 3.** Adsorption isotherms of caffeine on ZnO nanostructures: (**a**) ZnO sphere-caffeine; (**b**) ZnO rod-caffeine; (**c**) ZnO petal-caffeine (—Freundlich model, - - -Langmuir model).

Langmuir and Freundlich isotherms (Equations (2) and (3)) were used to fit the adsorption data and the plots are shown in Figure 3. The equilibrium adsorption capacity of ZnO nanostructures was calculated at different initial concentration of caffeine and the

isotherm parameters are presented in Table 3. The coefficient of regression ($R^2$) and chi-square values ($\chi^2$) were also tabulated for the assessment of goodness-of-fit for these models. We note that although both these models describe the experimental data well, yet Langmuir model provides a better fit since the correlation coefficient ($R^2$) values for Langmuir adsorption isotherm is higher than that for Freundlich isotherm. Langmuir isotherm is based on the assumption that the molecules to be adsorbed form a monolayer on the adsorbent surface and there is ideally no chemical reaction involved between the two (physisorption). In this study, the experimentally determined adsorption capacity is similar to the theoretical one, thus supporting the Langmuir isotherm.

**Table 3.** Isotherm parameters for Langmuir and Freundlich adsorption models.

| Sample | Langmuir Model | | | | Freundlich Model | | | |
|---|---|---|---|---|---|---|---|---|
| | $q_m$ (mg/g) | b (L/mg) | $R^2$ | $\chi^2$ | $1/n$ | $K_f$ (mg$^{1-1/n}$ (L) $^{1/n}$(g)$^{-1}$) | $R^2$ | $\chi^2$ |
| ZnO-sphere | 12.57 | 0.06 | 0.89 | 0.61 | 0.45 | 1.74 | 0.80 | 0.85 |
| ZnO-petal | 5.28 | 0.14 | 0.78 | 0.14 | 0.28 | 1.64 | 0.77 | 0.14 |
| ZnO-rod | 6.04 | 0.09 | 0.78 | 0.26 | 0.36 | 1.26 | 0.64 | 0.44 |

Freundlich isotherm is based on the assumption that there are chemical interactions involved between the molecules to be adsorbed and a heterogeneous adsorbing surface (chemisorption) leading to the formation of a multiple layers of adsorbing molecule. In our case, Freundlich isotherm fits the data as well, suggesting that caffeine molecules occupy heterogeneous adsorption sites on ZnO nanostructure surfaces and chemical interaction between caffeine molecules and ZnO may also contribute to the adsorption process. From Table 3, it can be seen that the values for b in Langmuir isotherm fall in between 0 and 1 which implies that the adsorption of caffeine on ZnO nanostructures is favorable [32]. Similarly, the parameter $1/n < 1$ in Freundlich isotherm also implies a favorable adsorption.

It was found that the maximum adsorption capacity ($Q_m$) for ZnO spheres (12.57 mg/g) was double compared to ZnO rods and petals. This difference in maximum adsorption capacities can be attributed to the different shapes, sizes and porosity of ZnO nanostructures.

### 2.3.2. Photocatalytic Experiments

We first tried to rule out the effect of photolysis from photocatalysis and for this purpose, we studied the degradation of caffeine in presence of 250 mW/m$^2$ visible light and absence of nanoparticles. The results shown in Supplementary Information (Figure S3) indicate that caffeine is only slightly affected by photolysis and thus we can exclude the effect of photolysis and perform photocatalytic study to describe the degradation. Next, we estimated the effect of different initial caffeine concentrations, ZnO concentration and morphologies and visible light intensities on the photocatalytic degradation of caffeine.

#### Nanoparticle Dosage

The effect of nanoparticle dosage on caffeine degradation was analyzed to find the lowest dosage at which the interaction of nanoparticles with caffeine can be maximized. We tested the degradation of caffeine at four different concentrations of ZnO (0.5, 1.0, 1.5 and 2.0 g/L) and it was seen that caffeine degraded faster when nanoparticle concentration was increased from 0.5 to 1.0 g/L for all morphologies of nanoparticles but it decreased at higher concentrations. This phenomenon can be explained by the tendency of nanoparticles to agglomerate at higher concentrations leading to a decrease in overall available surface area and an increase in diffusion time. Hence a nanoparticle dosage of 1.0 g/L was chosen for all subsequent experiments.

#### Initial Concentration of Caffeine and Nanoparticle Morphology

After the adsorption equilibrium was reached, we subjected the samples to 250 mW/cm$^2$ of simulated solar light and 1.0 g/L nanoparticle dosage for the photocatalytic experi-

ments. The photocatalytic data was fitted to pseudo-first order and pseudo-second order kinetic model (Equations (4) and (5)) as shown in Figure 4 and the value of rate constants were calculated from the above-mentioned equations using non-linear regression (Levenberg–Marquardt algorithm, $10^{-5}$ tolerance). From the $R^2$ values (shown in Table 4), it can be concluded that the experimental data can be suitably represented by both the kinetic models with high correlation coefficients. The rate constants for ZnO sphere falls with the rise in amount of caffeine. ZnO spheres exhibited the highest value of degradation rate constant at the lowest caffeine concentration of 10 ppm ($k_1$-1.323 $min^{-1}$, $k_2$-1.74 g/mg min). This is about 1.1 times ($k_1$ and $k_2$) higher than that achieved by petals whereas 1.6-($k_1$) and 2.2-($k_2$) times higher than the degradation achieved by ZnO rods. At highest concentration, ZnO spheres are 1.4-(first order) and 2.8-fold (second order) higher than ZnO petals whereas 2.6-(first order) and 2.5-fold (second order) higher compared to ZnO rods. Thus, the differences in the rate of caffeine degradation is more pronounced at higher concentrations irrespective of the morphology.

**Table 4.** Kinetic rate parameters for caffeine degradation by ZnO nanostructures (Experimental conditions: [ZnO] = 1.0 g $L^{-1}$, light intensity = 250 mW/$cm^2$).

| Sample | Parameter | Caffeine Concentration (ppm) | | | | | Light Intensity (mW/$cm^2$) | | |
|---|---|---|---|---|---|---|---|---|---|
| | | 10 | 20 | 30 | 40 | 50 | 50 | 150 | 250 |
| ZnO sphere | 0.1 $k_1$ * ($min^{-1}$) | 1.32 | 1.33 | 0.69 | 0.48 | 0.43 | 0.21 | 0.23 | 0.48 |
| | $R^2$ | 0.97 | 0.97 | 0.96 | 0.96 | 0.98 | 0.97 | 0.99 | 0.99 |
| | 0.01 $k_2$ * (g/mg min) | 1.74 | 0.97 | 0.26 | 0.16 | 0.10 | 0.07 | 0.06 | 0.09 |
| | $R^2$ | 0.94 | 0.99 | 0.99 | 0.98 | 0.99 | 0.97 | 0.99 | 0.99 |
| ZnO petal | 0.1 $k_1$ ($min^{-1}$) | 1.24 | 0.61 | 0.59 | 0.24 | 0.32 | 0.11 | 0.26 | 0.43 |
| | $R^2$ | 0.99 | 0.95 | 0.96 | 0.98 | 0.97 | 0.99 | 0.95 | 0.97 |
| | 0.01 $k_2$ (g/mg min) | 1.65 | 0.34 | 0.22 | 0.05 | 0.07 | 0.03 | 0.05 | 0.07 |
| | $R^2$ | 0.98 | 0.97 | 0.99 | 0.99 | 0.98 | 0.99 | 0.97 | 0.99 |
| ZnO rod | 0.1 $k_1$ ($min^{-1}$) | 0.81 | 0.35 | 0.15 | 0.12 | 0.16 | 0.10 | 0.03 | 0.12 |
| | $R^2$ | 0.96 | 0.98 | 0.98 | 0.98 | 0.99 | 0.96 | 0.99 | 0.99 |
| | 0.01 $k_2$ (g/mg min) | 0.77 | 0.15 | 0.03 | 0.02 | 0.04 | 0.02 | 0.01 | 0.02 |
| | $R^2$ | 0.96 | 0.99 | 0.98 | 0.99 | 0.99 | 0.96 | 0.99 | 0.99 |

* $k_1$ is pseudo-first order rate constant, and $k_2$ is pseudo-second order rate constant.

In terms of percentage degradation, it can be seen that ZnO spheres could degrade 72.5% of 30 ppm caffeine in 30 min and complete degradation was achieved in 120 min while the degradation rates for ZnO petals was 69% in 30 min and complete degradation in 135 min and rods was only 37% after 30 min and complete degradation in and complete degradation in 180 min. The corresponding degradation rates for different morphologies of ZnO can be listed as follows: ZnO spheres > ZnO petals > ZnO rods. Based on these findings, it could be perceived that the enhanced generation of ROS (Figure 2) in case of ZnO spheres might contribute to the increased photocatalytic activity.

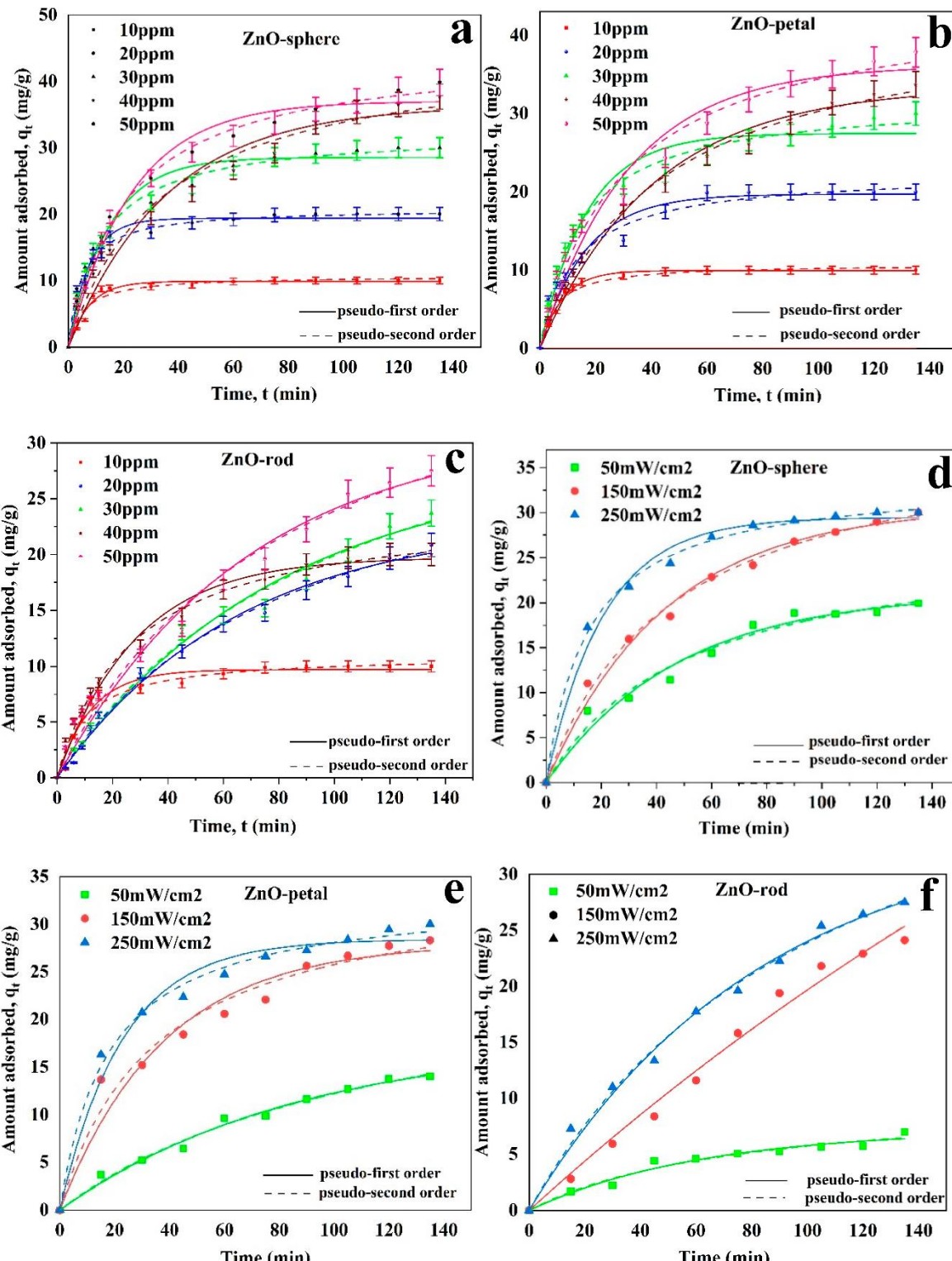

**Figure 4.** Kinetic model for the adsorption of caffeine on ZnO nanostructures at (**a**–**c**) various caffeine concentrations and (**d**–**f**) different light intensity. (—) and (- - -) represent pseudo-first-order and pseudo-second-order model, respectively.

Light Intensity

The degradation constant for caffeine increased with the increase in light intensity as a result of ROS generation and subsequent oxidation of caffeine (Figure 4d–f). The rate

constants ae presented in Table 4 and it was evident that a light intensity of 250 mW/cm$^2$ resulted in the highest rate of degradation.

When the solar light is incident on these nanostructures, photons are adsorbed on the surface of nanostructures causing electron-hole pair to form. These pairs reach the surface of nanostructures, the electrons (in conduction band) are scavenged by the $O_2$ molecules present in the medium leading to the generation of ROS ($\bullet O_2{}^-$) and the holes remain in the valance band. The holes also contribute to the generation of other ROS species (such as $\bullet OH$) by reacting with water molecules. These radicals react with the caffeine molecules, subsequently degrading it into harmless products such as $CO_2$ and $H_2O$.

### 2.4. Antibacterial Activity

#### 2.4.1. Minimum Inhibitory Concentration (MIC)

The minimum inhibitory concentration (MIC) of nanostructures for *E. coli* is listed in Table 5. MIC values for different nanostructures (for both *S.aureus* and *E. coli*) under consideration is as follows: ZnO sphere < ZnO rod < ZnO petal. Maximum value of MIC among all the nanostructures (90 µg/mL) was chosen for all subsequent experiments.

**Table 5.** MIC values for different morphologies of ZnO against *E. coli*.

| Sample | MIC Value (µg/mL) | Reference |
|--------|-------------------|-----------|
| ZnO-sphere | 60 | [33,34] |
|  | 78 | [14] |
|  | 65 | This study |
| ZnO-petals | 25 | [34,35] |
|  | 5 | [35,36] |
|  | 72 | This study |
| ZnO-rods | 2 | [36,37] |
|  | 512 | [37,38] |
|  | 90 | This study |

#### 2.4.2. Zone of Inhibition (Disc Diffusion Assay)

The antibacterial property of ZnO nanostructures was assessed against *E. coli* by standard disc diffusion assay. The diameter of zone of inhibitions obtained after overnight incubation with nanostructures (different concentrations) was measured and shown in Figure 5a,b. Among the different shapes of ZnO, ZnO spheres showed the best activity whereas ZnO petals and rods produced almost similar zone of inhibitions.

#### 2.4.3. Cell Viability Assay (CFU/mL)

To quantitatively assess the interaction of nanostructures with bacteria, the antibacterial tests were carried out in by suspending nanostructures (90 µg/mL) in liquid media contaning bacteria under constant shaking in dark at 37 °C. The number of viable bacteria were noted by counting the number of colonies (CFU/mL) and shown in Figure 5c. It was found that ZnO sphere and petals reduced the growth of bacteria to nearly 2.0 log CFU and 2.8 log CFU within 12 h, respectively. However, ZnO rods showed a much lower antibacterial activity of 5.0 log CFU reduction after 12 h treatment time. Thus among the different shapes of ZnO, ZnO spheres showed the highest reduction in cell viability followed by ZnO petals and ZnO rods.

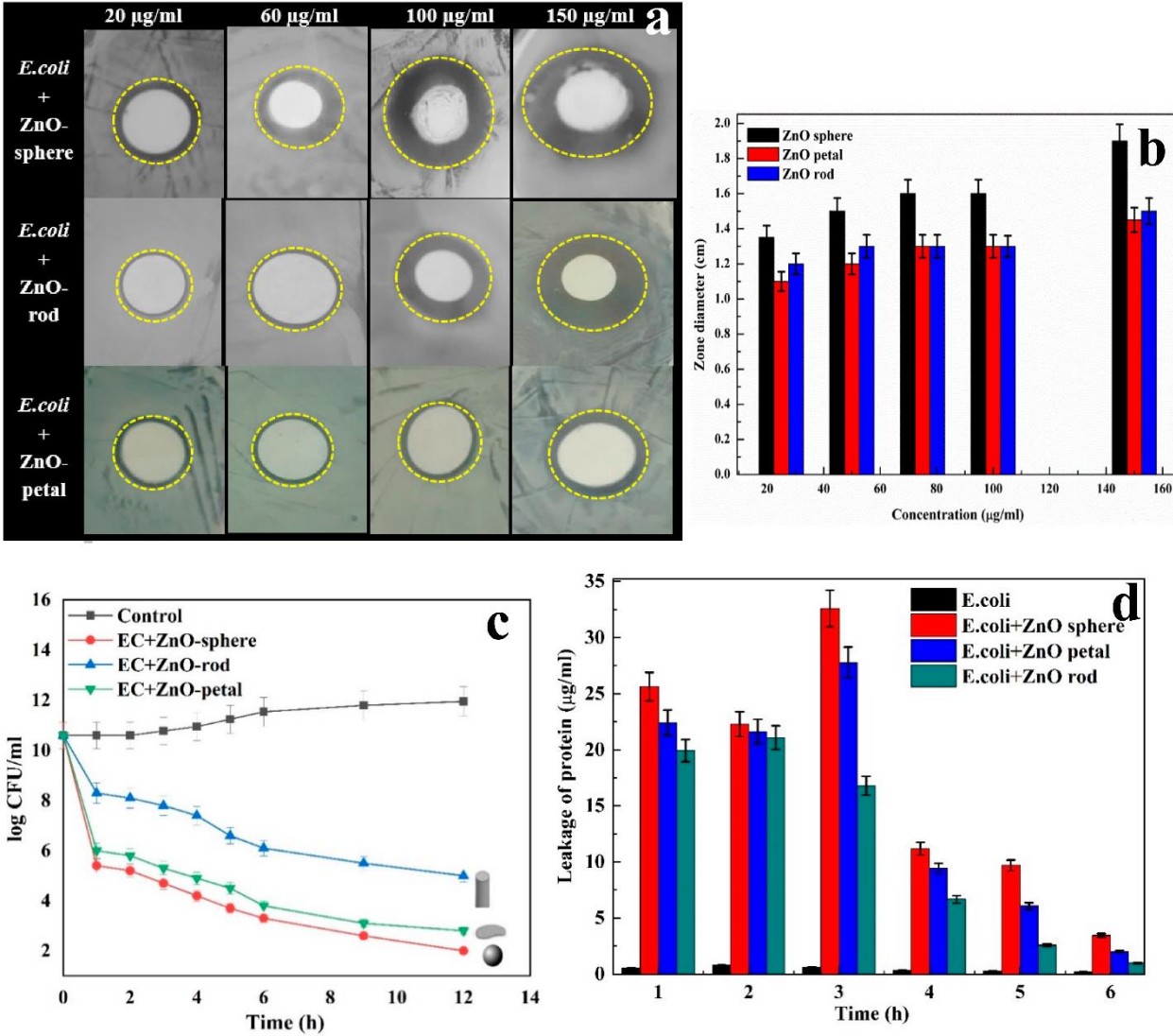

**Figure 5.** (**a**,**b**) Zone of inhibition (cm) formed in disc diffusion assay against *E. coli* for different concentration of nanostructures (20, 60, 100, 150 μg/mL) (**c**) Cell viability assay, and (**d**) Protein leakage analysis for *E. coli* treated with different morphologies of ZnO nanostructures.

### 2.4.4. Protein Leakage Analysis

In order to further confirm the rupture of membrane and leakage of cellular contents, the concentration of protein leaked out of the cell was monitored with time. Higher protein leakge is associated with a greated degree of membrane disruption. Leakage of proteins was detected in the medium containing nanostructure-exposed cells whereas almost negligible protein leakage was observed for untreated *E. coli* (Figure 5d). Maximum protein leakage was observed after 3 h for ZnO spheres and petals (32.59 and 22.41 μg/mL, respectively) and 2 h for rods (21.08 μg/mL). The permeability of cell membranes increase after proteins are leaked out in the soution which leads to oxidative stress in the cells [39].

### 2.4.5. Imaging

The FESEM images and EDS spectra of control and nanoparticle treated *E. coli* are shown in Figure 6a,c,e,g,i–k. Before the antibacterial treatment, *E. coli* had a well-defined rod shape and an integrated cell wall structure. On exposure to nanostructures, the original shape of bacteria was damaged with holes and ruptured cell membrane and the nanostructures were seen to be in close proximity with the cells. Drainage of cell contents from *E. coli*

was also evident. *E. coli* following nanoparticle exposure revealed the presence of ZnO petals near the cells (Figure 6g). EDS spectra of bacteria-nanoparticle system indicates ZnO peaks which indicates ZnO around the damaged bacteria specimen.

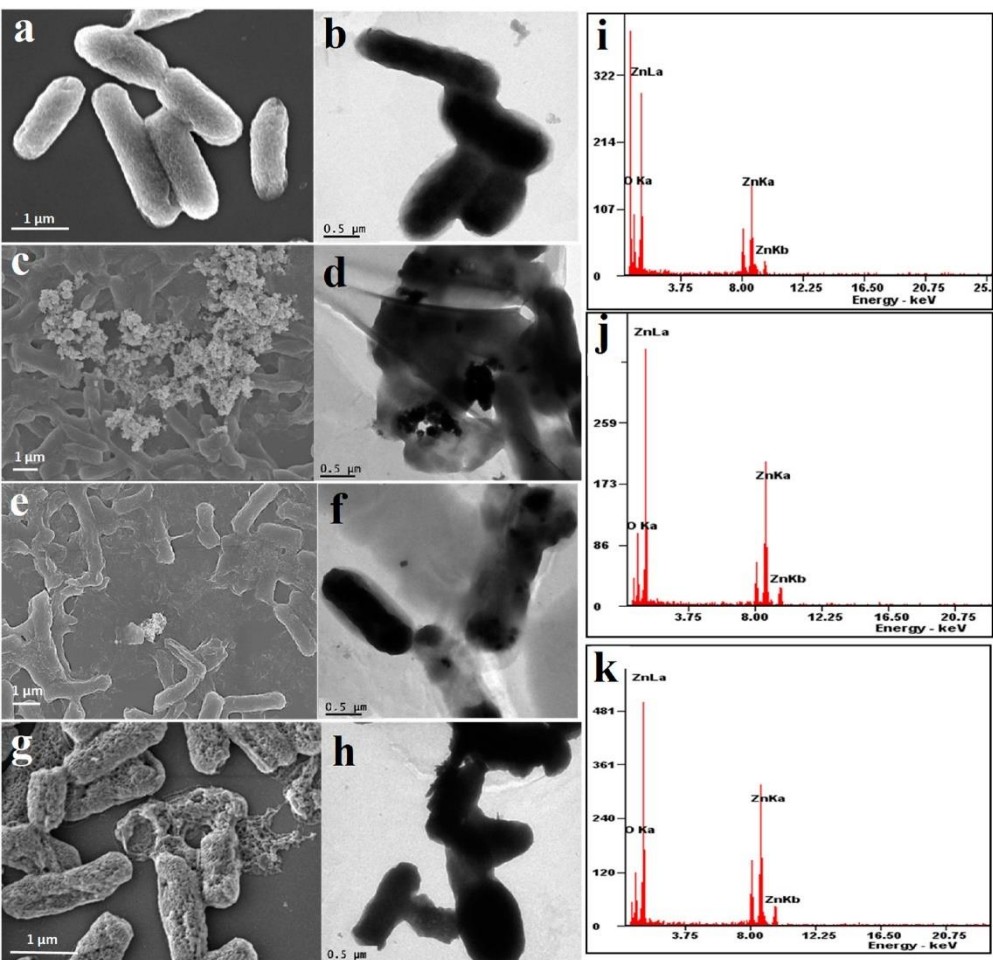

**Figure 6.** Scanning electron microscope images (**a**,**c**,**e**,**g**), Transmission electron microscope images (**b**,**d**,**f**,**h**) and EDS spectra for (**i**–**k**) for (**a**,**b**) Control (**c**,**d**,**i**) *E. coli* treated with ZnO sphere (**e**,**f**,**j**) *E. coli* treated with ZnO petals (**g**,**h**,**k**) *E. coli* treated with ZnO rod.

To further visualize the direct interaction between nanostructures and bacteria, TEM imaging experiments were performed. ZnO spheres penetrated the cell walls and internalize the cells leading to cell death (Figure 6e). ZnO petals did not internalize the cells (Figure 6f) and the antibacterial activity can be attributed to physical attrition. No internalization was seen in case of ZnO rods as well (Figure 6h), which implies that the sharp edges of ZnO rods pierced the cells and physical attrition led to the cell damage. Apart from physical attrition and cell internalization, ROS generation and $Zn^{2+}$ toxicity are also responsible for the antibacterial activity of nanostructures. All the structures were found to adhere to the bacteria cell walls but except for ZnO spheres, no internalization was observed for other particles. Thus, cell wall damage and cytoplasm leakage for rods and petals was caused by other mechanisms such as physical attrition, ROS generation and cation release. The small size of ZnO spheres enable them to penetrate the bacterial cell resulting in cell death. The ROS and cations react with the intracellular contents of the bacteria and inactivate the cell. Studies have shown that ROS released from bigger nanostructures interacts with the cell membrane of these bacteria and oxidize the phospholipid contents of the membrane [13,40]. However, for smaller structures, the internalized contents are more damaged compared to the cell membrane. Thus, the rigidity of cell

membrane, generation of holes and pits on the surface and leakage of internal contents are mostly determined by the morphology of nanostructures. Another theory that may find its applicability in this case is the abrasiveness of different nanostructures. ZnO rods have rough edges and sharp ends that causes mechanical damage to the membranes (physical attrition and piercing action) creating holes and pits causing localized leakage of cell contents.

All the antibacterial experiments with ZnO nanostructures were conducted in dark, suggesting that the cell membrane distortion by physical interaction with nanostructures and reaction of cell contents with cations might be a bigger contributor to cell damage compared to ROS [41,42].

Being a chemical process, there is no directly visible effect of nanoparticle shape on caffeine degradation and it is mostly governed by the ROS and $Zn^{2+}$ released by nanostructures. On the contrary, shape of nanoparticles were visually evident to play a major role in *E. coli* disinfection since physical interaction of nanoparticles with bacteria is one of the ways that lead to cell death. For example, nanoparticles with sharp edges have a different mechanism of killing bacteria as compared to smooth surfaces. However, in addition to shape, we cannot also rule out the role of surface area in the photocatalytic and antibacterial activity of ZnO nanoparticles. For example, nanoparticles with larger surface area (ZnO sphere) were found to generate more ROS and $Zn^{2+}$ as compared to the ones with lower surface area (ZnO petals and rods), thus affecting caffeine degradation in different ways. Additionally, smaller nanoparticles could internalize the bacterial cell and distort the cellular contents as compared to larger nanostructures that were limited to outer cell wall damage. Thus, the effect of surface area must also be considered in addition to shape to gain a clear understanding of the photocatalytic and antibacterial activity of these nanostructures.

## 3. Experimental Details

### 3.1. Materials

All the materials used in this work were of analytical grade and used without further purification. Zinc nitrate hexahydrate [99%] was purchased from JT Baker Chemicals, Phillipsburg, NJ, USA and sodium hydroxide [NaOH] was obtained from Caledon Laboratories Ltd., Georgetown, ON, Canada. Caffeine and polyethylene glycol (Kollisolv PEG E 400) were procured from Sigma Aldrich, Oakville, ON, Canada. Pure agar and Luria Bertani (LB) media were provided by HI MEDIA Laboratories, Bengaluru, Karnataka, India. Bacteria *Escherichia coli* (NCIM 2137) was purchased from NCIM, NCL Pune, Pune, Maharashtra, India.

### 3.2. Method

#### 3.2.1. Synthesis of Nanoparticles

Different morphologies of ZnO nanoparticles were obtained by the hydrolysis reaction of zinc nitrate hexahydrate $Zn(NO_3)_2 \cdot 6H_2O$ in different solvents. The details of the synthesis procedure can be found elsewhere [14]. Briefly, 0.1 M $Zn(NO_3)_2 \cdot 6H_2O$ in 100 mL solvent (either PEG400 or water or toluene) was kept under magnetic stirring followed by the addition of 2 M NaOH. After 45 min, the nanoparticles were collected by ultracentrifugation, washing, overnight drying for 12 h and calcining at 300 °C.

#### 3.2.2. Characterization of Nanomaterials

The synthesized nanoparticles were characterized using scanning electron microscope (SEM, Hitachi, Tokyo, Japan) for surface morphology. Nanoparticles were coated with platinum before imaging to avoid the charge accumulation near the surface of the particles. The shape and size of the nanomaterials were assessed using a JEOL High-Resolution Transmission Electron Microscope (HRTEM, Tokyo, Japan). BET analysis was performed for the estimation of porosity and specific surface area of nanomaterials using a Quantachrome ASiQwin-Automated Gas Sorption Data Acquisition and Reduction instrument

(outgassing time: 1.5 h, temperature: 270 °C, Boynton Beach, FL, USA) with nitrogen as the adsorption gas. The phase composition of the samples was determined using a Panalytical High Resolution XRD-I, PW 3040/60 (Chennai, Tamil Nadu, India). The 2-theta angle was from 30° to 80° using a CuK$\alpha$ radiation ($\lambda$ = 1.5418 Å). Zeta potential of particles was measured by Zetasizer Nano-ZS-90 (Malvern Instruments, Malvern, UK) with the scattering angle in the range of 90° to 25°. Nanoparticles (10 µg/mL) were ultrasonically dispersed in distilled water and measured under automatic mode. Dissolution of metal ions in water was studied for all the nanoparticles at a concentration of 90µg/mL. Aliquots from the supernatant of the suspension were collected and nanoparticles were separated by centrifugation. The supernatant was analyzed by PinAAcle 900 H atomic absorption spectrometer (AAS), Waltham, MA, USA. The standard deviation of triplicate measurements has been reported for all the experiments. The superoxide radicals ($\bullet O_2^-$) from ZnO nanoparticles were measured using nitroblue tetrazolium (NBT) assay [14]. One mM NBT was added to 100 mL of methanol (4%) and nanoparticles (90 µg/mL) under magnetic stirring, regular samples were withdrawn and UV-vis spectra was recorded from 200–400 nm. The hydroxyl radicals (OH$\bullet$) were estimated by measuring the reaction product of terephthalic acid in solution. Briefly, a solution of 0.5 mM of terephthalic acid was prepared in NaOH and ZnO nanoparticles (90 µg/mL) were ultrasonically dispersed in the solution. Samples were withdrawn, centrifuged and the excitation fluorescence spectra was taken at 315 nm using.

### 3.2.3. Photocatalytic Degradation of Caffeine
Adsorption Study

The adsorption of caffeine on different morphology of ZnO nanoparticles was studied using UV–vis spectrophotometer (UV-3600, Shimadzu Scientific Instruments, Columbia, MD, USA) and the spectra were recorded in a range of 250–350 nm. Batch adsorption experiments were conducted with 1 g/L of ZnO nanoparticles with different caffeine concentrations (~10, 20, 30, 40, 50 ppm) under shaking in dark at 150 rpm for 2 h. The adsorption studies were performed to determine the time required to reach equilibrium and to calculate the adsorption coefficients. The equilibrium adsorption capacity of caffeine on nanoparticles ($q_e$, mg/g) was calculated by Equation (1):

$$q_e = (C_0 - C_e) \times \frac{V}{W} \tag{1}$$

where $C_0$ and $C_e$ are the initial and equilibrium concentrations of caffeine (ppm), respectively, $V$ is the volume of caffeine solution (L) and $W$ is the weight of nanoparticles used (g).

Adsorption isotherms are fundamental for describing the adsorption capability of adsorbents. The equilibrium adsorption isotherms were developed by plotting the equilibrium concentration of caffeine and its corresponding uptake. The Langmuir isotherm is based on the assumption of monolayer adsorption of adsorbate on the adsorbent surface, and is given by the following expression:

$$q_e = \frac{b q_m C_e}{1 + b C_e} \tag{2}$$

where, $C_e$ (mg/L) and $q_e$ (mg/g) represent the concentration and amount of adsorbed material, respectively and $q_m$ (mg/g) and $b$ (L/mg) are the maximum adsorption capacity and Langmuir constant, respectively. The Freundlich isotherm is valid for adsorption on a heterogeneous surface and is given by

$$q_e = K_F (C_e)^{\frac{1}{n}} \tag{3}$$

where, $K_F$ (mg/g) and $n$ are capacity and adsorption favorability, respectively.

Kinetic Study

In a typical kinetic study, the important variables that affect the photocatalytic process are photocatalyst concentration, initial concentration of pollutant and light intensity. Batch experiments were conducted using nanoparticle/caffeine suspension of required concentration at neutral pH. The solution was kept under shaking at 150 rpm and the samples were analyzed in UV–vis spectrophotometer at regular intervals. The aliquots were filtered through a 0.45 μm filter prior to the absorbance measurements. A solar simulator (SS1 KW, ScienceTech, London, ON, Canada) equipped with a 1000 W Xe arc lamp and an Air-Mass (AM) 1.5 G filter was used to irradiate the sample with visible light.

The optimum amount of ZnO nanoparticles required for caffeine degradation was determined first. For this, four different concentrations of nanoparticles (0.5, 1, 1.5, 2 g/L). To investigate the effect of light intensity on caffeine degradation four different light intensities (50, 150 and 250 mW/cm$^2$) were used at optimum dosage of nanoparticles (1 g/L) and 30 ppm caffeine concentration. The initial concentration of caffeine (10, 20, 30, 40, 50 ppm) was subsequently varied at 250 mW/cm$^2$ light intensity (which provides highest degradation). All the experiments were carried out under constant magnetic stirring at room temperature. From the adsorption studies, it was seen that the adsorption equilibrium was reached after 1.5 h and accordingly, the photocatalytic experiments were carried out after 1.5 h of dark reaction.

The kinetics of caffeine adsorption by nanoparticles was investigated using a pseudo-first-order and pseudo-second-order kinetics, given by the following equations:

$$q_t = q_e(1 - \exp(k_1 t)) \tag{4}$$

$$q_t = \frac{k_2 q_e^2 t}{1 + k_2 q_e t} \tag{5}$$

where, $q_t$ and $q_e$ are the adsorption capacities of nanoparticles (mg/g) at time $t$ and equilibrium respectively and $k_1$ (min$^{-1}$) and $k_2$ (g/mg min) are the pseudo-first-order and pseudo-second order rate constants, respectively.

3.2.4. Bacterial Toxicity Assessment

*E. coli* was incubated overnight in Luria Bertani (LB) broth at 37 °C under constant shaking (120 rpm). The CFU/mL of bacteria was adjusted to 10$^8$ for all the experiments unless otherwise stated.

Minimum Inhibitory Concentration (MIC)

The MIC value is the minimum concentration of nanomaterials that prevents any visible development of pathogen colonies. To obtain the MIC values, overnight grown cultures of *E. coli* and nanostructures (10–150 μg/mL) were mixed and incubated for 24 h. The optical density (OD) was recorded at 590 nm and the MIC value was reported as the one that inhibited the growth of 99% bacteria. The MIC value obtained from experiments (90 μg/mL) will be used for all subsequent experiments unless otherwise stated.

Zone of Inhibition

For the disc diffusion assay, different concentrations of nanoparticles (25, 50, 75, 100 and 150 μg/mL) were ultrasonically coated on paper discs (8 mm). Ten μL bacteria (10$^8$ CFU/mL) was spread on solidified agar plates and the paper disk was positioned above the bacteria coated agar and kept at 37 °C for 12 h. The zone diameter (mm) after incubation was measured and a mean value of triplicate experiments was noted.

Cell Viability Assay (CFU/mL)

Change in bacterial CFU with time in absence and presence of nanoparticles provides an estimate of the antibacterial strength of nanoparticles. Bacteria was incubated at 37 °C with nanoparticles and the CFU was recorded at intermediate times for a duration of

12 h. The CFU value obtained from the average of three experiments falling within 95% confidence interval were recorded.

Protein Leakage Analysis (Bradford Assay)

The protein released by nanoparticle treated deformed cells was quantified by Bradford assay [43]. Briefly, cells were washed and re-dispersed in PBS along with nanoparticles and incubated under shaking at 120 rpm and 30 °C. Aliquots of sample (1 mL) were withdrawn every hour, centrifuged and a solution of 200 μL supernatant and 800 μL Bradford's reagent was prepared and incubated in dark for 10 min. The absorbance of the samples was measured at 595 nm with the help of a spectrophotometer.

Imaging of Bacteria-Nanoparticle Interaction

Damage to the bacteria cell morphology before and after exposure to nanoparticles was observed in SEM. Samples were platinum coated prior to the SEM analysis. Bacteria treated with nanoparticles for 10 h were dropped on a coverslip, kept in 2.5% glutaraldehyde solution for 12 h at 4 °C and then dehydrated using ethanol. And the samples were vacuum dried prior to imaging. For the TEM imaging, a drop of untreated and treated bacteria was dropped on a carbon coated copper grid, dried and imaged.

## 4. Conclusions

Morphology-controlled ZnO were successfully synthesized by wet chemical process. The nature of solvent used for synthesis had a significant impact on the size and direction of crystal growth. Electron microscopy images showed that sphere, petal and rod-like ZnO were formed in PEG400, water and toluene, respectively. These nanostructures were studied for caffeine degradation and *E. coli* disinfection. The ROS and $Zn^{2+}$ ions generated by these nanostructures were mainly responsible for the nanoparticle activity. The ROS consisted of $O_2^-$ and OH• while $H_2O_2$ was found to be insufficient to show any photocatalytic or antibacterial activity. These nanostructures were very effective for the degradation of caffeine using solar light which has a relatively low operating cost compared to UV photocatalytic processes. The nanostructures were also effective in killing 99% of bacteria at low concentrations. However, some questions still remain unanswered and further studies are needed to determine the intermediate products of caffeine degradation by different nanostructures. Future studies should also focus on how photocatalytic and antibacterial activity are affected by other parameters such as water chemistry, pH, temperature, etc. and the toxicity of different nanostructures in environment.

**Supplementary Materials:** The supplementary materials are available online at https://www.mdpi.com/2073-4344/11/1/63/s1.

**Author Contributions:** Conceptualization: S.T., S.N., A.K.R.; methodology: S.T.; resources: S.N. and A.K.R.; writing—original draft preparation: S.T.; writing—review and editing: S.T. and A.K.R.; supervision: S.N. and A.K.R. All authors have read and agreed to the published version of the manuscript.

**Funding:** This research did not receive any specific grant from funding agencies in the public, commercial, or not-for-profit sectors.

**Institutional Review Board Statement:** Not applicable.

**Informed Consent Statement:** Not applicable.

**Data Availability Statement:** The data presented in this study are available on request from the corresponding author. The data are not publicly available due to privacy restrictions.

**Acknowledgments:** We thank the research facilities of IIT Kharagpur, India and Western University, Canada for providing the required necessary equipment. The first author would like to express her hearty gratitude to Malini Ghosh for her help in data analysis.

**Conflicts of Interest:** The authors declare no conflict of interest.

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
