# Peer review of "Morphology-Controlled Synthesis of ZnO Nanostructures for Caffeine Degradation and Escherichia coli Inactivation in Water"

_catalysts, doi:10.3390/catal11010063_

Round 1
Reviewer 1 Report
Comments to the Author
The manuscript by Thakur et al. describes the preparation morphology-controlled ZnO nanoparticles by simple metal salt hydrolysis using different solvents. The work is aiming at achieving the optimum photocatalytic and antibacterial activities. The authors demonstrated three types of morphologies and reported photocatalytic degradation activity variations for Caffeine and Escherichia coli inactivation. Both the photoctalytic and antibacterial experimental parameters are optimized and structural characterizations were performed and well presented. However, the authors should address the following issues before the article is considered for publications
- The authors used different solvents (water, PEG, toluene) and obtained three types of morphologies, however the reader miss how these solvents influence the morphologies. I suppose there are interplays between rate of hydrolysis and seeding and growth. One cannot also rule out the role of solvent templating. Please add more details on the influence of solvents in the final morphologies of the nanoparticles.
- In the XRD patterns (Fig 1h), the (101) reflex for the ZnO rod visibly shifted to low diffraction angle, clarify this observation.
- The authors presented two adsorption isotherms (Langmuir vs Freundlich) and the data fits reasonably for both. However, the authors decide to use the Langmuir isotherm “as the data results a better fit”. But this argument may lead to unreliable results as the two models fundamentally differ. So the data should be treated carefully and reasoning should base on surface / molecule interactions.
- As the authors described, the reactive oxygen species (ROS) and Zn2+ ions are responsible for the activity of the ZnO nanostructures. However, ROS include superoxide radicals, hydroxyl radicals and holes. While authors investigate the superoxide radicals the other two are left out. How do we know the superoxide radicals are the dominant ROS? Or the activity is a combination of all three?
- The authors suggest that the Zn2+ cations also play a role in the activity; a control experiment by dosing a specific Zn2+ without the ZnO nanostructure would be interesting. These will strength the suggestion how ZnO petals and rods antibacterial activities work.
- The electron microscopy images presented in Figure 6 together with the EDX data are of poor quality and difficult to read the scale bar and the legends inside the images, please provide a better quality image.
Author Response
Reviewer 1
Comment 1. The authors used different solvents (water, PEG, toluene) and obtained three types of morphologies, however the reader miss how these solvents influence the morphologies. I suppose there are interplays between rate of hydrolysis and seeding and growth. One cannot also rule out the role of solvent templating. Please add more details on the influence of solvents in the final morphologies of the nanoparticles.
Response: We thank the reviewer for pointing this out. A separate section (Section 2.2, line number: 135-148) named “Mechanism of morphology change with the choice of solvent” has been included to incorporate the influence of solvents in the final morphologies of the nanoparticles.
Comment 2. In the XRD patterns (Fig 1h), the (101) reflex for the ZnO rod visibly shifted to low diffraction angle, clarify this observation.
Response: The reason for (101) peak shift has been clarified and the following statements have been included: “It can be seen that the (101) peak for ZnO nanorods is shifted towards lower diffraction angle. This may be attributed to the change in growth direction from (101) to (002) thereby reducing the strain along (101) direction.” (line number: 96-98)
Comment 3. The authors presented two adsorption isotherms (Langmuir vs Freundlich) and the data fits reasonably for both. However, the authors decide to use the Langmuir isotherm “as the data results a better fit”. But this argument may lead to unreliable results as the two models fundamentally differ. So the data should be treated carefully and reasoning should base on surface / molecule interactions.
Response: We have included additional parameter comparisons that help us to distinguish and choose between the two models. By comparing the experimental and theoretical adsorption capacities, we found that Langmuir isotherm is a better fit for our system. However, the possibility that Freundlich isotherm also fits our system cannot be ruled out because of the heterogeneous nature of nanoparticles and the adsorption sites on the surface. (line number: 172-179)
Comment 4. As the authors described, the reactive oxygen species (ROS) and Zn2+ ions are responsible for the activity of the ZnO nanostructures. However, ROS include superoxide radicals, hydroxyl radicals and holes. While authors investigate the superoxide radicals the other two are left out. How do we know the superoxide radicals are the dominant ROS? Or the activity is a combination of all three?
Response: We thank the reviewer for this question. We have included the results for the generation of hydroxyl radicals by ZnO nanostructures. However, holes that are formed in the valance band have a tendency to react with electron donor (like hydroxyl ions) to form hydroxyl radicals (•OH) [1]. Thus, we have not separately estimated the number of holes generated by nanoparticles as the determination of hydroxyl radicals would suffice for holes as well. (line number: 123-133)
Ref: [1] Yang Li, Wen Zhang, Junfeng Niu, and Yongsheng Chen, Mechanism of Photogenerated Reactive Oxygen Species and Correlation with the Antibacterial Properties of Engineered Metal-Oxide Nanoparticles, ACS Nano 2012, 6, 6, 5164–5173, https://doi.org/10.1021/nn300934k
Comment 5. The authors suggest that the Zn2+ cations also play a role in the activity; a control experiment by dosing a specific Zn2+ without the ZnO nanostructure would be interesting. These will strength the suggestion how ZnO petals and rods antibacterial activities work.
Response: While we understand that control experiment by dosing a specific Zn2+ without the ZnO nanostructure would strengthen our hypothesis, it is not possible for the authors to perform experiment at this time. However, we will keep this important point in mind and perform the control experiment for our future studies.
Comment 6. The electron microscopy images presented in Figure 6 together with the EDX data are of poor quality and difficult to read the scale bar and the legends inside the images, please provide a better quality image.
Response: Figure 6 has been modified for improved quality and readability.
Reviewer 2 Report
The manuscript by Ray and co-authors shows the influence of a photocatalyst morphology on the photocatalytic and antibacterial activities.
Results are well presented, as well as the conclusions drawn by the authors.
The only remark that could be done is the following: Authors should report the intermediates of the photocatalytic degradation of caffeine.
Author Response
Reviewer 2
Comment 1. Authors should report the intermediates of the photocatalytic degradation of caffeine.
Response: The authors thank the reviewer for his/her suggestion. But unfortunately, it is not possible for the authors to perform experiment at this time. However, we will keep this important point in mind and estimate the intermediate products for our future studies.
Reviewer 3 Report
This manuscript article by Thakur and coauthors compares the degradation efficiency and toxicology of different ZnO morphologies. Clearly, the material fabrication is not the main point of this manuscript, instead the connection between morphology and photocatalytic properties are presented. The research design is appropriate and the introduction is very nicely written. The motivation is clearly presented. However, the manuscript is quite long with unnecessary data and a stronger focus on the main findings would help the reader not to loose interest while reading.The presentation of the kinetic studies and few other figures are of poor quality.The abstract is a bit long, as well. The use of paragraphs is not appropriate because the abstract should embody a single paragraph. Overall I am not convinced that this manuscript is appropriate for the Journal catalysis - both in regard to scope and presentation quality. In-depth revision would be required but I recommend the authors to invest the time and effort. Thus, I suggest major revisions before a final acceptance could be recommended.
Additional comments/questions:
1) The introduction is excellently written with the motivation nicely explained. However, before the sentence “This study aims at… “ a paragraph break would be necessary.
2) Is it truly morphology controlled, or is it simply a matter of available surface area? The respective differences in surface area of the morphologies should be compared and included in the discussion.
3) The placement of the subfigure marks (a,b,c, etc.) in Fig. 4 is suboptimal.
4) The error bars in Fig 2c are barely visible (line width too small). Similar issues can be found in Fig. 4abc. Please revise.
5) The difference between the Langmuir and the Freundlich model is not clear.
6) The term “time kill assay” appears to be easily misunderstood.
7) Figure 6 has several issues and appears very messy. The SEM, TEM and EDX plots are affected by limited resolution, artefacts and are strange offsets. These presentation issues need to be resolved. The EDX plots cannot be read at all.
8) The separate discussions section is superfluous. Discussion should be included in the results section. Also, the conclusions are a bit repetitive to the results and discussion sections. The main findings should be subsumed in a brief but motivated way with a critical view on open questions and motivate future studies. Overall, the conclusions are weakly written.
9) The references have flaws, e.g. DOI of Ref 39 is incorrect.
Author Response
Reviewer 3
Comment 1. The introduction is excellently written with the motivation nicely explained. However, before the sentence “This study aims at… “ a paragraph break would be necessary.
Response: A paragraph break before the sentence “This study aims at… “ has been included as per the reviewer’s suggestion.
Comment 2. Is it truly morphology controlled, or is it simply a matter of available surface area? The respective differences in surface area of the morphologies should be compared and included in the discussion.
Response: We thank the reviewer for pointing this out. A paragraph explaining the role of morphology and surface area in the photocatalytic and antibacterial activity of nanoparticles has been included. (line number: 325-336)
Comment 3. The placement of the subfigure marks (a,b,c, etc.) in Fig. 4 is suboptimal.
Response: The subfigure marks in Fig. 4 has been corrected.
Comment 4. The error bars in Fig 2c are barely visible (line width too small). Similar issues can be found in Fig. 4abc. Please revise.
Response: The line width of error bars in Figure 2c and 4 abc has been increased for a better visibility.
Comment 5. The difference between the Langmuir and the Freundlich model is not clear.
Response: We have included a few more sentences elaborating on the difference between Langmuir and the Freundlich model. (line number: 172-179)
Comment 6. The term “time kill assay” appears to be easily misunderstood.
Response: We have changed the term “time kill assay” to “Cell viability assay”. (line number: 436)
Comment 7. Figure 6 has several issues and appears very messy. The SEM, TEM and EDX plots are affected by limited resolution, artefacts and are strange offsets. These presentation issues need to be resolved. The EDX plots cannot be read at all.
Response: Figure 6 has been modified for improved quality and readability.
Comment 8. The separate discussions section is superfluous. Discussion should be included in the results section. Also, the conclusions are a bit repetitive to the results and discussion sections. The main findings should be subsumed in a brief but motivated way with a critical view on open questions and motivate future studies. Overall, the conclusions are weakly written.
Response: The discussion section has been merged with the results and the conclusions have been rewritten incorporating the reviewer’s suggestion.
Comment 9. The references have flaws, e.g. DOI of Ref 39 is incorrect.
Response: We apologize for the flawed references. The references have now been checked and corrected where required.
Round 2
Reviewer 3 Report
The authors have revised their contribution in part. However few issues still remain, for which I need to recommend further revision to improve the quality of presentation.
1) Repeating comment 3, the placement of subfigure marks in Fig 4 is still suboptimal. Labels are cutoff and partially hidden.
2) Repeating comment 5, the difference between the Langmuir and the Freundlich model is not clear. Please explain the assumption of each model in particular how these are of signifiance fort he presented data.
3) Repeating comment 7, Figure 6 has still several issues and appears still very messy. The SEM, TEM and EDX plots are affected by limited resolution, artefacts, strange insets, hard to read scale bars, and are strange offsets, etc. These presentation issues need to be resolved. The EDX plots cannot be read and need tob e replotted properly.
Author Response
Comment 1. Repeating comment 3, the placement of subfigure marks in Fig 4 is still suboptimal. Labels are cutoff and partially hidden.
Response: We have corrected the labels in Fig 4.
Comment 2. Repeating comment 5, the difference between the Langmuir and the Freundlich model is not clear. Please explain the assumption of each model in particular how these are of signifiance fort he presented data.
Response: We have included the assumptions of each model and their significance in the results section (line number: 169-181).
Comment 3. Repeating comment 7, Figure 6 has still several issues and appears still very messy. The SEM, TEM and EDX plots are affected by limited resolution, artefacts, strange insets, hard to read scale bars, and are strange offsets, etc. These presentation issues need to be resolved. The EDX plots cannot be read and need to be replotted properly.
Response: Figure 6 has been revised for a better presentation. The readability of EDX plats has also been improved.